# First Report of *Phodopus sungorus* Papillomavirus Type 1 Infection in Roborovski Hamsters (*Phodopus roborovskii*)

**DOI:** 10.3390/v13050739

**Published:** 2021-04-23

**Authors:** Grega Gimpelj Domjanič, Lea Hošnjak, Maja M. Lunar, Lucijan Skubic, Tomaž Mark Zorec, Joško Račnik, Blaž Cigler, Mario Poljak

**Affiliations:** 1Institute of Microbiology and Immunology, Faculty of Medicine, University of Ljubljana, 1000 Ljubljana, Slovenia; grega.gimpelj-domjanic@mf.uni-lj.si (G.G.D.); lea.hosnjak@mf.uni-lj.si (L.H.); maja.lunar@mf.uni-lj.si (M.M.L.); lucijan.skubic@mf.uni-lj.si (L.S.); tomaz-mark.zorec@mf.uni-lj.si (T.M.Z.); 2Institute of Poultry, Birds, Small Mammals and Reptiles, Veterinary Faculty, University of Ljubljana, 1000 Ljubljana, Slovenia; josko.racnik@vf.uni-lj.si; 3Miklavžin Veterinary Center, Lakotence 7a, 1000 Ljubljana, Slovenia; bazi.cigler@gmail.com

**Keywords:** papillomavirus, *Phodopus sungorus* papillomavirus type 1 (PsuPV1), *Phodopus roborovskii*, molecular analysis, phylogenetic analysis, evolution, viral cross-species transmission

## Abstract

Papillomaviruses (PVs) are considered highly species-specific with cospeciation as the main driving force in their evolution. However, a recent increase in the available PV genome sequences has revealed inconsistencies in virus–host phylogenies, which could be explained by adaptive radiation, recombination, host-switching events and a broad PV host range. Unfortunately, with a relatively low number of animal PVs characterized, understanding these incongruities remains elusive. To improve knowledge of biology and the spread of animal PV, we collected 60 swabs of the anogenital and head and neck regions from a healthy colony of 30 Roborovski hamsters (*Phodopus roborovskii*) and detected PVs in 44/60 (73.3%) hamster samples. This is the first report of PV infection in Roborovski hamsters. Moreover, Phodopus sungorus papillomavirus type 1 (PsuPV1), previously characterized in Siberian hamsters (*Phodopus sungorus*), was the only PV detected in Roborovski hamsters. In addition, after a detailed literature search, review and summary of published evidence and construction of a tanglegram linking the cladograms of PVs and their hosts, our findings were discussed in the context of available knowledge on PVs described in at least two different host species.

## 1. Introduction

Papillomaviruses (PVs) comprise a diverse group of small non-enveloped DNA viruses that are known to infect either cutaneous or mucosal stratified epithelia of various vertebrate species. Most often, PVs are part of the normal host microbiota, whereas specific PV types can cause a variety of different neoplastic lesions. PV-induced lesions frequently regress spontaneously, but in some cases, they can progress to a more severe disease, such as cancer [1,2,3,4,5]. Based on the official PV nomenclature, PV’s taxonomic ranking is set according to the similarity of the highly conserved full-length L1 genes at the 60% nucleotide sequence similarity for genera, 70% for species and 90% for types [6,7]. With 224 human (HPV) and 215 animal PV types currently characterized and officially recognized, *Papillomaviridae* is one of the largest viral families infecting vertebrates [8,9]. In addition, complete PV genome sequences annotated with clinical and biochemical data that are readily available make PVs an attractive research model for study of the evolution of pathogen-host interactions [10,11,12].

Although substantial data are available, the evolutionary forces that drive PV diversity are yet to be fully understood. Predominantly, PVs are considered highly species-specific, with virus–host cospeciation as the main driving force of their evolution [13,14,15,16]. According to Fahrenholz’s rule, this requires a clear congruence between viral and host phylogenies. However, an increasing number of PV genomic sequences has exposed several incongruities in tree topologies, and some evidence has even suggested that only one-third of evolutionary events can be associated with virus–host co-divergence [17,18]. These inconsistencies may be explained by different forces of evolution such as adaptive radiation, recombination, host-switching events and broad PV host range [17,19,20]. Namely, several studies have shown that the same PV types can infect different hosts; for example, various bovine (BPV1, BPV2, BPV13 and BPV14), gull (GuPV2), duck (DuPV3) and bat (Eptesicus serotinus papillomaviruses, EsPV2 and EsPV3) PVs were found in at least two different host species [19,21,22,23]. Interestingly, overlapping sets of non-human primate PVs (Macaca mulata papillomavirus, MmPV1 and Macaca fascicularis papillomaviruses, MfPV3, MfPV8 and MfPV11) were also detected in different host species [24]. A broad PV host range could be associated with specific cellular tropism of the virus. For example, *Deltapapillomaviruses* (*Delta*-PVs) known for their broad host range, were shown to have different cellular tropism compared to other more species-specific PVs, predominantly infecting epithelial cells, as they are able to infect both the epithelium and the underlying connective tissue [25,26,27]. Extrapolating these findings to humans could suggest that there are more routes of acquiring PV infection than currently suggested, and that domestic animals in particular, with close contacts with humans, could represent a potential source of transmission. Thus, we believe that a better understanding of host-switching events and the molecular mechanisms behind viral infections is an essential step toward mitigating the burden of PVs on society.

To clarify the evolution of PVs, a more balanced approach in taxonomic sampling should be adopted, as more than half of the currently recognized PV genomes correspond to HPVs, leading to limited knowledge of PVs in other vertebrate species. The practicality of rodents as model systems in studying the biology and pathogenesis of these viruses makes detailed identification and characterization of rodent PVs even more vital. Currently there are only 16 recognized rodent PVs, clustering in four different PV genera: *Iotapapillomavirus* (*Iota*-PV), *Pipapillomavirus* (*Pi*-PV), *Sigmapapillomavirus* (*Sigma*-PV) and *Dyosigmapapillomavirus* (*Dyosigma*-PV). Of these, four rodent PV types were identified in the brown rat (*Rattus norvegicus*; RnPV1-4), two in the North American porcupine (*Erethizon dorsatum*; EdPV1 and -2) and one in Chevrier’s field mouse (*Apodemus chevrieri*; AchePV1), the wood mouse (*Apodemus sylvaticus*; AsPV1), the southern multimammate mouse (*Mastomys coucha*; McPV2), the Natal multimammate mouse (*Mastomys natalensis*; MnPV1), the harvest mouse (*Micromys minutus*; MmiPV1), the house mouse (*Mus musculus*; MmuPV1), the deer mouse (*Peromyscus maniculatus*; PmPV1), the Syrian hamster (*Mesocricetus auratus*; MaPV1), the Siberian hamster (*Phodopus sungorus*; PsuPV1) and the North American beaver (*Castor canadensis*; CcanPV1), respectively [8,28].

To date, prevalence of PsuPV1 was only studied in Siberian hamsters that together with Roborovski (*Phodopus roborovskii*) and Campbell’s dwarf (*Phodopus campbelli*) hamsters comprise the genus *Phodopus* [29]. To improve knowledge of biology and the spread of hamster PVs, we sampled and analyzed mucosal swabs of the anogenital and head and neck regions of a healthy colony of Roborovski hamsters, using the quantitative PsuPV1 real-time (qPCR) and broad range *Pipapillomavirus* (*Pi*-PV PCR) PCRs. In addition, after a detailed literature search, review and summary of published evidence and construction of a tanglegram linking the cladograms of PVs and their hosts, our findings were extensively discussed in the context of available knowledge on PV types described in at least two different animal species.

## 2. Materials and Methods

### 2.1. Sample Collection

In 2017, during routine health checks of 30 Roborovski hamsters belonging to the same colony, paired swab samples of the oral (lips, mouth and cheek pouches) and anogenital (vagina or penis/preputium and anus) regions were collected from each animal by experienced veterinary staff, using sterile EUROTUBO Collection Swabs (Deltalab, Barcelona, Spain). To avoid potential sample-to-sample cross-contamination, strict hygiene and disinfection measures were applied throughout the sampling procedure. The samples were stored in the original tubes at −80 °C until further analyses.

### 2.2. Nucleic Acid Extraction

Total DNA of collected samples was extracted using a QIAamp DNA Mini Kit (Qiagen, Hilden, Germany), following the manufacturer’s instructions. During the DNA extraction process, standard precautions and strict laboratory practices were in place to minimize the risk of cross-contaminations [30]. Subsequently, the concentration of the isolated DNA was measured using a NanoDrop 2000c spectrophotometer (Thermo Fisher Scientific, Waltham, MA, USA). In all samples, the integrity of extracted DNA and absence of PCR inhibitors were verified by qPCR, targeting a 65 base-pair (bp) fragment of the tubulin gene, using previously published primers (Tub-F: 5′-TCCTCCACTGGTACACAGGC-3′; Tub-R: 5′-CATGTTGCTCTCAGCCTCGG-3′) and a hydrolysis probe (5′-FAM AGGGCATGGACGAGATGGAGTTCA-BBQ-3′), as already described [31,32].

### 2.3. Detection of Phodopus Sungorus Papillomavirus Type 1 and Pipapillomaviruses

DNA isolates were initially screened with the PsuPV1 type-specific qPCR primer set (APV12-L1-RT-F: 5′-GATCCCAAGCAGACTCAAATG-3′ and APV12-L1-RT-R: 5′-ACCTGCATTAATTTGGTTACAAGG-3′), targeting the 100 bp fragment of the PsuPV1 L1 gene. The PsuPV1 qPCR test was performed using a QuantiTect SYBR Green PCR kit on a LightCycler 1.5 Instrument (Roche Diagnostics, Mannheim, Germany). Samples were considered PsuPV1-positive when showing specific melting peaks at around 77.5 °C. Furthermore, PsuPV1-negative samples were subjected to a highly sensitive broad-range *Pi*-PV PCR assay that allows the amplification of a 330 bp L1 gene fragment of all currently recognized *Pi*-PVs.

All *Pi*-PV PCR products were analyzed using 2% agarose gel electrophoresis. In addition, 16 randomly selected PsuPV1 qPCR and all five *Pi*-PV PCR–positive products were Sanger sequenced on an Applied Biosystems 3500 Genetic Analyzer (Thermo Fisher Scientific, Waltham, MA, USA) automated sequencing instrument and analyzed using the BLAST algorithm [33]. All analyses were carried out in line with our previously published protocols [31].

### 2.4. Hamster Species Identification

Hamster species were determined based on the morphological characteristics of the animals studied and the evaluation of the nucleotide sequences consisting of the partial cytochrome b and threonine-specific tRNA (tRNA-Thr) housekeeping genes of 22 randomly selected tubulin-positive DNA isolates, as described previously [34]. The obtained nucleotide sequences of the partial cytochrome b and tRNA-Thr genes were compared with the corresponding parts of the Roborovski hamster genome sequence from the NCBI GenBank database (accession no. KU885975) using the BLAST algorithm [33]; only sequences with more than 98% similarities to the reference sequence were considered to cluster in this hamster species.

### 2.5. Sequencing of the Complete Genome of Phodopus Sungorus Papillomavirus Type 1

Three randomly selected PsuPV1 L1 gene–positive samples were first subjected to rolling circle amplification (RCA) using an Illustra TempliPhi 100 Amplification Kit (GE Healthcare Life Sciences, Little Chalfont, UK), following the manufacturer’s instructions. The PsuPV1 type-specific primer set (PsuPV1_LNGL1-F: 5′-CGTTGCTAAACAAACTAGGTGAC-3′ and PsuPV1_LNGL1-R: 5′-GATGTCCGGTTGTCCCTAC-3′), targeting the PsuPV1 L1 gene, was subsequently used to amplify their complete viral genomes, with the help of inverted long-range PCR in combination with a Platinum *Taq* DNA Polymerase High Fidelity Kit (Invitrogen, Carlsbad, CA, USA), as described previously [31].

Sanger sequencing of the PCR products obtained was performed by a primer-walking strategy, using 22 sequencing primers, as already described [31]. Sequences were constructed using the Vector NTI Advance v11.5.4 (Thermo Fisher Scientific, Waltham, MA, USA) and BioEdit Sequence Alignment Editor v7.2.6.1 (Ibis Therapeutics, Carlsbad, CA, USA) [35] software packages and compared with the PsuPV1 reference sequence (GenBank acc. no. HG939559), using the BLAST algorithm [33].

### 2.6. Phylogenetic Analysis of Papillomaviruses and Their Hosts

Phylogenetic analysis of PVs was based on their complete L1 gene sequences. In addition, phylogenetic analysis of the complete cytochrome b gene sequences (routinely used for determining phylogenetic relationships between organisms within the same families and genera) of corresponding host animals was carried out to study their cospeciation [32,36]. All PV L1 and cytochrome b gene sequences were obtained through the Papillomavirus Episteme (PaVE) [8] and GenBank sequence databases, respectively (GenBank acc. no. available in Appendix A). Multiple nucleotide sequence alignments were obtained using the MUSCLE algorithm available in the MEGA v7.0.26 software package (Institute for Genomics and Evolutionary Medicine, Temple University, Philadelphia, PA, USA) [37]. To construct phylogenetic trees, the PhyML v3.0 software package was used with the Akaike information criterion for automatic model selection and 1000 bootstrap replicates to evaluate the significance of the tree topology. Finally, a tanglegram was created and edited using the Dendroscope-3 v3.7.3 and Inkscape v1.0.1 software packages, respectively [38,39].

### 2.7. Papillomavirus Cross-Species Transmission Literature Search

Previously published studies on PV cross-species transmissions were searched for in the NCBI PubMed database under the following search terms: “interspecies transmission” or “cross-species transmission” and “papillomavirus”.

## 3. Results

Using the tubulin qPCR, it was confirmed that all 60 collected Roborovski hamster DNA isolates were adequate for downstream analyses. Moreover, by sequencing the partial cytochrome b and tRNA-Thr housekeeping genes [29] it was validated that all animals included in this study were indeed Roborovski hamsters.

As shown in Table 1, using the PsuPV1 qPCR the presence of the PsuPV1 was detected in 39/60 (65.5%) hamster samples. The average cycle threshold (Ct) values of the qPsuPV1 PCR- positive samples from the oral and anogenital regions were 35.5 (range: 32.8–40.0) and 34.6 (range: 25.9–40.0), respectively. The specificity of the PsuPV1 qPCR was additionally verified by Sanger sequencing of 16 randomly selected PsuPV1 qPCR–positive samples (16/16, 100%). Subsequently, all PsuPV1 qPCR–negative samples were further tested with the *Pi*-PV PCR, with which 5/21 (23.8%) samples were successfully amplified and additionally characterized as PsuPV1 isolates. Consequently, a total of 44/60 (73.3%) samples, representing 27/30 (90.0%) hamsters, were positive for the PsuPV1. Specifically, PsuPV1 was confirmed in 76.7% and 70.0% of oral and anogenital mucosa samples, respectively (Table 1).

By sex of the animals, 25/28 (89.3%) PsuPV1-positive samples (14 from oral and 11 from anogenital mucosa) were identified in female Roborovski hamsters and 19/32 (59.4%) (nine from oral and 10 from anogenital mucosa) in male Roborovski hamsters. In particular, PsuPV1 was identified in all 14 females and 13/16 (81.3%) males, of which 10 (71.4%) female and six (37.5%) male hamsters were PsuPV1-positive in both oral and anogenital mucosa samples.

Following the initial amplification and sequencing of the PsuPV1 L1 gene fragments, complete PsuPV1 genomes were sequenced from three randomly selected samples collected from the oral (APV475 and APV480) and anogenital mucosa (APV516) of Roborovski hamsters. The genomic sequences obtained were almost identical to the reference nucleotide sequence of the PsuPV1 genome (GenBank acc. no. HG939559) and only the specified single-nucleotide variants (SNVs) were identified at the following nucleotide positions: APV475 (GenBank acc. no. MW602287)–826 (E7: C/T–A138V), 2736 (E2: C/T–A41V), 6046 (L1: C/T–A176V) and 6390 (L1: C/T–H291Y); APV480 (GenBank acc. no. MW602286)–826 (E7: C/T–A138V), 2736 (E2: C/T–A41V), 3478 (E2: C/G), 6046 (L1: C/T–A176V) and 6390 (L1: C/T–H291Y); APV516 (GenBank acc. no. MW602285)–826 (E7: C/T–A138V), 2736 (E2: C/T–A41V), 6303 (L1: G/A–G262R) and 6390 (L1: C/T–H291Y).

These results show that PsuPV1 can infect two different species of hamsters, Siberian and Roborovski hamsters. In addition, these results were placed in the broader context of other reported PV types infecting more than one host species. As shown in Table 2, there are currently 16 animal PV types described in at least two different animal species.

As shown in Figure 1, PV variants found in the Roborovski hamster population cluster together with PsuPV1, first identified in Siberian hamsters. Correspondingly, phylogeny based on the complete cytochrome b gene sequences of the hosts’ species shows a monophyletic relationship of the Siberian and the Roborovski hamsters, and thus suggests their common ancestry [29]. Importantly, the tanglegram indicates that PV cross-species transmissions occur almost exclusively between closely related animal species. Namely, most of the PV cross-species transmissions occurred between hosts belonging to the same order and, with the exception of bovine (BPV1, BPV2, BPV8, BPV13 and BPV14) and sheep (Ovis aries papillomavirus, OaPV2) PV types, only between members of the same genus. As mentioned, OaPV2 interspecies transmission was described in the order Artiodactyla (i.e., even-toed ungulates) between two different families of bovids (*Bovidae*) and suids (*Suidae*). Even more interestingly, bovine PV types were found on more than one occasion to cross the species barrier among higher taxonomic ranks of the animal order. In most cases, these host jumps occurred between the closely related orders Perissodactyla (i.e., odd-toed ungulates) and Artiodactyla, which together comprise a diverse clade of ungulates. As of today, only the interspecies transmission of BPV14 was described outside the clade Ungulata, between the orders Artiodactyla and Carnivora.

## 4. Discussion

In recent years, the increase in newly annotated PV genomes has provided a more detailed understanding of the evolutionary relationships between PVs while revealing a number of incongruities in virus–host phylogenies [51]. In particular, strong representation of human PVs and a disproportionally limited understanding of animal PVs stand in the way of fully explaining the evolutionary forces that drive viral diversification and explaining the discrepancies between virus–host phylogenies. This study aimed to expand the repertoire of animal PVs and help build more balanced insight into the evolution of PVs. Because of their usefulness as model systems in studying the biology of PVs, we specifically focused on the rodent population. Namely, rodents are becoming increasingly important as preclinical infection model systems, aiding in understanding the natural history of HPV infection, disease progression, its prevention and potential treatment [28]. Specifically, one of the more important rodent PVs, MmuPV1, was isolated from the laboratory strain of mice and proved extremely useful in studying PV infections in laboratory settings [52]. Some exciting research that was made possible using mice models includes diversified vaccine development, new insight into ultraviolet radiation-induced tumorigenesis associated with PV infection and improved understanding of oncogenic proteins [53,54,55,56]. We believe that more detailed identification and characterization of rodent PVs will lead to further applications of these models and specifically aid in understanding virus–host coevolution and improving our knowledge of virus–host interactions in general.

In our previously published work [31], we focused on the population of Siberian hamsters and described a full-genome characterization, the prevalence and the clinical significance of the novel PsuPV1 virus. PsuPV1 was detected in anogenital and oral swab samples from Siberian hamsters, where it mainly causes asymptomatic infections. In addition, occasionally it can also be etiologically associated with the development of oral squamous cell carcinoma. To further evaluate the distribution of PV infections in hamsters, we analyzed 30 healthy Roborovski hamsters by collecting paired samples of anogenital and oral mucosa regions from individual animals. Altogether, the prevalence of PsuPV1 in the Roborovski hamster population analyzed was 90.0% (27/30), with more cases detected among female hamsters (14/14; 100%) compared to the male hamster population (13/16; 81.3%), however gender difference was not statistically significant. Similarly, the proportion of PsuPV1-positive animals in both oral and anogenital mucosa samples was higher in female (10/14; 71.4%) than male (6/16; 37.5%) hamsters. Using the PsuPV1 qPCR, PsuPV1 was detected in 66.7% (20/30) of oral and 63.3% (19/30) of anogenital swab samples, suggesting a balanced distribution of PsuPV1 in both anatomical sites. In addition, PsuPV1 was also detected in 30.0% (3/10) of anogenital and 18.2% (2/11) of oral PsuPV1 qPCR–negative samples, using the broad range *Pi*-PV PCR. Because the PsuPV1 qPCR has a high analytical sensitivity [31], additional PsuPV1-positive samples detected by the *Pi*-PV PCR could be explained by mismatches in qPCR primer/probe annealing sites that were surpassed by the degenerative primers designed for broad-range detection of all *Pi*-PVs. Contrary to the Siberian hamster population, no oral squamous cell carcinoma and no oral precancerous lesions were detected among 27 PV-infected Roborovski hamsters, although all three complete PsuPV1 genomic sequences obtained in this study (GenBank acc. no. APV475: MW602287; APV480: MW602286; APV516: MW602285) carried SNVs previously reported in an isolate from the oral squamous cell carcinoma of a Siberian hamster (GenBank acc. no. LM653111) at the following nucleotide positions: 826 (E7: C/T–A138V), 2736 (E2: C/T–A41V) and 6390 (L1: C/T–H291Y). Nevertheless, because these mutations were found outside the major functional domains of corresponding viral genomic regions, their effect on viral lifecycle and potential increased risk of carcinogenesis remains unknown.

This study shows that PsuPV1 is able to infect both Roborovski and Siberian hamster populations. However, it remains unknown whether the PsuPV1 was already present in its common hamster ancestor and subsequently coevolved with the respective hosts after the split into Roborovski and Siberian hamsters or if its presence in both hamster species is the result of an interspecies viral transmission. Based on current understanding of animal PVs, the coevolution hypothesis seems more likely. Unfortunately, present knowledge on the PsuPV1 cellular tropism is not adequate to make any significant conclusions [31]. In view of that, it would be interesting to investigate the potential presence of PsuPV1 in other hamster species, particularly closely related third representative of the genus *Phodopus*, Campbell’s dwarf hamster, as well to study its cellular tropism for different cells and tissues [29]. Our review of the available literature on PV cross-species transmission shows that there are currently at least 16 different animal PVs known to infect different host species. Despite the fact that these PV types belong to six different viral genera, the majority of PV cross-species transmission cases are found within the genus *Delta-PV*. Specifically, BPV1 was described in nine and BPV2 in seven different animal species, representing 16/30 (53.3%) of all cases described. A broad host range of *Delta*-PVs could be associated with their specific cellular tropism, as in addition to epithelial cells, they can also infect fibroblasts of the connective tissue, while most other PVs are epitheliotropic and more strictly species-specific [25,26,27].

Most frequently, cross-species transmissions of PVs occur between closely related host species and are most common between hosts from the same genus, and only a few sporadic cases have been reported in animals within different taxonomic ranks of family and order. However, based on the high diversity of animal species, only a limited number were studied for PV prevalence, thus it is difficult to conclude whether these taxonomic restraints are true across the entire *Papillomaviridae* family. In particular, PV tropism, host ecology and behavior could play an important role in determining viral transmission patterns and evolution. Because closely related animals usually share similar ecological niches, the probability of chance encounters increases and a virus becomes more likely to host jump. For instance, in BPVs many transmission events were described between different farm animals sharing the same living spaces [40,41]. More notably, BPV14 was isolated from cows and cats, which, despite their different evolutionary past, share a similar habitat, where their close physical contact is not uncommon [48]. Interestingly, strict virus–host coevolution is also challenged by the existence of several polyphyletic PV lineages within the same host [18]. Here, this is particularly well observed among bat, cattle and human PVs that are scattered throughout the PV phylogenetic tree but are found within the same hosts. Apart from the data included in our phylogeny, polyphyletic lineages of PVs have also been described in various primates, sheep, horses, dolphins, cats, dogs and porpoises [19]. This indicates that intra-host divergence and niche adaptation also play an important role in virus evolution. The importance of these adaptations was recently well established in primates, including humans, where strong phylogeny-tropism is associated with determining viral genomics, pathogenicity, carcinogenicity and host specificity [57,58]. This shows that understanding virus–host interactions and coevolution can lead to better disease prevention and improved knowledge of the viral pathogenicity.

Despite accumulating evidence showing less stable species barriers, it is still not completely clear what are the key mechanisms determining the PV ability to migrate between different host species. Some studies included in our literature review only reported PV E5 gene or LCR region sequence identities, which, based on the current nomenclature, do not suffice for the characterization of detected PVs as the same PV type and provide less reliable evidence of PV cross-species transmissions. Likewise, most of the studies describing PV cross-species transmissions fail to show whether the reported cases are the result of the common ancestry of animal species, or whether PV types are able to circulate within the population of different species without any restrictions.

In conclusion, we identified, sequenced and annotated the entire genome of the PsuPV1 previously characterized in Siberian hamsters in a novel species: the Roborovski hamster. As a result, we have shown that PsuPV1 is able to infect both Siberian and Roborovski hamsters, indicating the high value of these hamster species as novel rodent models for PV research in general. In addition, our literature search on PV types infecting more than one host species and virus-host phylogeny comparison showed that PVs cross-infect mostly highly related hosts, clustering within the same genus. Unfortunately, it has not yet been determined what role host ecology and behavior play in this and to what extent this applies to the entire *Papillomaviridae* family. To clarify this, a more systematic approach in sampling animal PVs should be adopted. With better understanding of PV evolution, we will gain important knowledge that will aid in future research on viral pathogenesis and disease prevention, and provide unique insight into pathogen-host interactions.

## Figures and Tables

**Figure 1 viruses-13-00739-f001:**
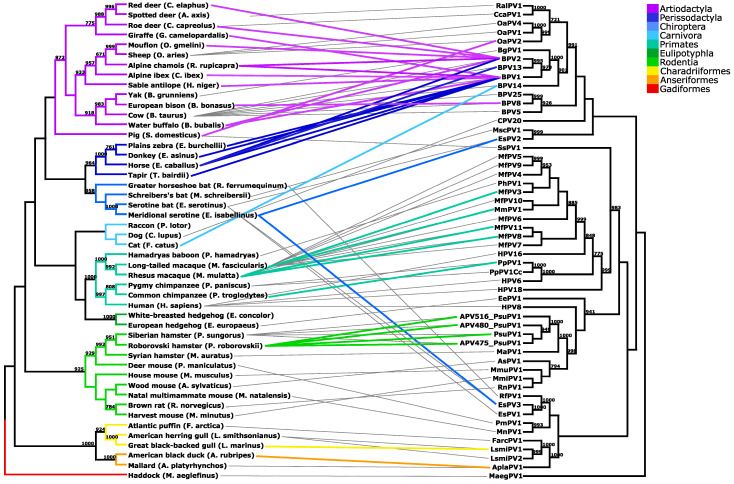
Tanglegram linking the cladograms of papillomaviruses (PVs) and their hosts. Different clade colors indicate the respective host order. Gray lines connect PVs with their original hosts, and colored lines link the corresponding PV cross-Scheme 700. are shown.

**Table 1 viruses-13-00739-t001:** Sample analysis overview.

Sample Type	No. Samples	PsuPV1 qPCR–Positive	*Pi*-PV PCR–Positive	Total PsuPV1-Positive
Oral mucosa	30	20/30 (66.7%)	3/10 (30.0%)	23/30 (76.7%)
Anogenital mucosa	30	19/30 (63.3%)	2/11 (18.2%)	21/30 (70.0%)

PsuPV1: Phodopus sungorus papillomavirus type 1, qPCR: quantitative real-time PCR, *Pi*-PV PCR: *Pipapillomavirus* PCR.

**Table 2 viruses-13-00739-t002:** Overview of animal papillomaviruses described in at least two different host species.

PV Type.	PV Genus	Sample Origin	Detection Method	Host A	Host B	Percent Identity of the L1 Gene	Study
BPV1	*Delta*-PV	Sarcoid tumors	PCR, Sequencing	*Bos taurus* (cow)	*Equus asinus* (common donkey)	96–98%	[40]
		Skin swabs	PCR, Sequencing		*Equus caballus* (horse)	*	[41]
		Neoplastic sarcoid tissue	PCR, Sequencing		*Tapirus bairdii* (tapir)	99% (E5 identity reported)	[42]
		Cutaneous and vulvar fibropapilloma	PCR, Sequencing		*Bubalus bubalis* (water buffalo)	largely homologous (LCR identity reported)	[25]
		Lesions	PCR, Sequencing		*Giraffa camelopardalis* (giraffe), *Hippotragus niger* (sable antelope)	97% (E2 and E5 identity reported)	[43]
		Neoplastic tissue	PCR, Sequencing		*Equus burchellii* (plains zebra)	98% (244-bp E5 identity reported)	[44]
		Proliferative lesions	PCR, Sequencing		*Rupicapra rupicapra* (chamois), *Capra ibex* (Alpine ibex)	99.2–99.9%	[45]
BPV2	*Delta*-PV	Bladder sections	PCR, Sequencing	*Bos taurus* (cow)	*Bubalus bubalis* (water buffalo)	100%	[46]
		Skin swabs	PCR, Sequencing		*Equus caballus* (horse)	*	[41]
		Skin from coronary band and labial mucosa	PCR, Sequencing		*Cervus elaphus* (red deer), *Rupicapra rupicapra* (chamois), *Capreolus capreolus* (roe deer), *Ovis gmelini* (mouflon)	99–100% (E5 identity reported)	[22]
		Proliferative lesions	PCR, Sequencing		*Rupicapra rupicapra* (chamois)	100%	[45]
BPV8 (BAPV2)	*Delta*-PV	Papillomatous lesions	PCR, Sequencing	*Bos taurus* (cow)	*Bison bonasus* (European bison)	100%	[47]
BPV13	*Delta*-PV	Sarcoid lesions	PCR, Sequencing	*Bos taurus* (cow)	*Equus caballus* (horse)	99–100%	[21]
BPV14	*Delta*-PV	Feline sarcoid lesions	PCR, Sequencing	*Bos taurus* (cow)	*Felis catus* (cat)	100%	[48]
OaPV2	*Delta*-PV	Oral sarcoid-like mass	PCR, Sequencing	*Ovis aries* (sheep)	*Sus domesticus* (domestic pig)	99.6–99.7%	[49]
MmPV1 (RhPV1)	*Alpha-PV*	Exfoliated cervical cytology samples	PCR, Sequencing	*Macaca mulatta* (rhesus macaque)	*Macaca fascicularis* (long-tailed macaque)	97.1–97.7%	[24]
MfPV3 (RhPVd)	*Delta*-PV			*Macaca fascicularis* (long-tailed macaque)	*Macaca mulatta* (rhesus macaque)	93.9%	
MfPV8 (RhPVa)	*Delta*-PV			*Macaca fascicularis* (long-tailed macaque)	*Macaca mulatta* (rhesus macaque)	94.6%	
MfPV11 (RhPVb)	*Delta*-PV			*Macaca fascicularis* (long-tailed macaque)	*Macaca mulatta* (rhesus macaque)	93.2%	
PpPV1	*Delta*-PV		PCR, Sequencing	*Pan paniscus* (pygmy chimpanzee)	*Pan troglodytes* (common chimpanzee)	93.2%	[50]
PsuPV1	*Pi-PV*	Oropharyngeal and anogenital swabs	PCR, Sequencing	*Phodopus sungorus* (Siberian hamster)	*Phodopus roborovskii* (Roborovski hamster)	99.9%	Current study
AplaPV1 (DuPV3)	Unclassified	Oropharyngeal and cloacal swabs	PCR, Sequencing	*Anas platyrhynchos* (mallard)	*Anas rubripes* (American black duck)	95.8%	[23]
LsmiPV1 (GuPV2)	Unclassified			*Larus smithsonianus* (American herring gull)	*Larus marinus* (great black-backed gull)	98.2%	
EsPV2 (EserPV2)	*Dyoomega-PV*	Oropharyngeal, anogenital swabs and hair bulbs	PCR, Sequencing	*Eptesicus serotinus* (serotine bat)	*Eptesicus isabellinus* (meridional serotine)	**	[19]
EsPV3 (EserPV3)	*Dyopsi-PV*			*Eptesicus serotinus* (serotine bat)	*Eptesicus isabellinus* (meridional serotine)	**	

Host A refers to the animal host in which the PV was first described and Host B to all hosts with subsequent identifications of the same PV type. * Different identities within the same host reported. ** No identity reported. PV: papillomavirus; BPV: bovine papillomavirus; OaPV: Ovis aries papillomavirus; MmPV: Macaca mulata papillomavirus; RhPV: Rhesus papillomavirus; MfPV: Macaca fascicularis papillomavirus; PpPV: Pan paniscus papillomavirus; PsuPV: Phodopus sungorus papillomavirus; AplaPV: Anas platyrhynchos papillomavirus; DuPV: Duck papillomavirus; LsmiPV1: Larus smithsonianus papillomavirus; GuPV: Gull papillomavirus; EsPV/EserPV: Eptesicus serotinus papillomavirus.

## Data Availability

The authors confirm that the data supporting the findings of this study are openly available in the GenBank database at https://www.ncbi.nlm.nih.gov/genbank/, accessed on 26 March 2021, under accession numbers MW602285 (PsuPV1 genome, isolate APV516), MW602286 (PsuPV1 genome, isolate APV480) and MW602287 (PsuPV1 genome, isolate APV475).

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
