# Peer review of "First Report of Phodopus sungorus Papillomavirus Type 1 Infection in Roborovski Hamsters (Phodopus roborovskii)"

_viruses, 2021, doi:10.3390/v13050739_

Round 1

Reviewer 1 Report

This is a very interesting paper on the discovery of PsuPV1 in a population of Roborovski hamsters. The study is very well described, the paper is well presented and written. Yet, I have a problem with the interpretation of results and of the scientific merit of the work:

On basis of your findings, you built up a story line on the generally accepted species-specificity of PVs, arguing that your findings are indicative for the possibility that PVs may be far less species-specific than currently thought.

In principle, I fully agree that it is very important to question scientific dogmata and congratulate you for this critical thinking. However, I am not convinced that you drew the right conclusions in case of your data.

Hamsters originate from a common ancestor, at least in case of the genus Phodopus that represents the most ancient evolutionary split in the hamster subfamily. From your data, you cannot tell whether PsuPV1 was already present in the common ancestor or not. Hence, it remains unclear, whether the virus truly jumped species. And I think that this is how you could discuss the subject.

Second: As example of true cross-species infection by PVs, you mention delta-PVs, which is perfectly correct. Yet, delta-PVs also differ from other PVs regarding their cell tropism. It actually appears that strictly epitheliotropic PVs are also strictly species-specific, as also experimentally shown, for example for CRPV, COPV, BPV4, HPVs etc. In contrast, the few PVs that also, or predominantly,infect fibroblasts (mainly delta-PVs) also have a broader host range.

Hence it seems more than likely that the respective cell tropisms of a PVs determine the degree of host-specifity.

On these grounds, information on the cell tropism of PsuPV1 should be included in the introduction and findings discussed in this context - that would be really interesting.

Regarding the scientific merit of your findings, I would mainly emphasize the high value of P. roborovskii as novel rodent model for PV research.

Please see also document attached.

Reviewer 2 Report

Papillomavirus (PV) is one of the largest viral families infecting vertebrates, but with limited knowledge in non-human species. PV are known highly host-specific, but several studies have shown that the same PV types can infect different hosts. Regarding human papillomavirus (HPV), there is no pre-clinical animal model to study in extenso the relationship that HPV establishes with its host, given the restriction of the HPV replicative cycle to differentiated human keratinocytes. A better understanding of host-switching events is crucial to possibly design new pre-clinical animal models.

To improve knowledge of the biology of rodent PV, Grega Gimpelj Domjanic et al. analyzed swabs of anogenital and oral regions of 30 healthy Roborovski hamsters using both specific and broad-range PCRs for the detection of Phodopus sungorus papillomavirus type 1 (PsuPV1) and viruses belonging to Pipapillomavirus genera, respectively.

A total of 73.3% of the hamsters’ samples are positive for PsuPV1 PCR, from which L1 sequencing of 16 samples of them confirmed its identification. They concluded first that PsuPV1 can infect two different species of hamsters, Siberian and Roborovski hamsters, and that PsuPV1 can therefore cross the species barrier, demonstrating for the first time a PV interspecies transmission within the genus Phodopus. The Roborovski hamster PsuPV1 entire genome was sequenced and annotated. The authors provided an overview of cross-species transmission of animal papillomaviruses within different host species and a nice tanglegram to plot the comparisons.

The study is well conducted and the manuscript well-written. I have only a few questions, that are minor points and would suggest accepting this original study for publication after minor revision.  

Minor points:

  • It would be interesting to know the quantification (or the semi-quantification or the Ct values) of the qPsuPV1 PCR positive samples.
  • It would be interesting to know why authors used PsuPV1 PCR only and not MaPV1 or others. How was the choice of primers (i.e. PsuPV1 and Pipapillomavirus PCRs) made in this case?
  • Is the unique L1 sequencing relevant to distinct PV types? Why not using the full-length genome sequence? This point could be better discussed.
  • Why RfPV and EsPV are so different in the tanglegram?
